# Autophagy Modulation in Aggresome Formation: Emerging Implications and Treatments of Alzheimer’s Disease

**DOI:** 10.3390/biomedicines10051027

**Published:** 2022-04-29

**Authors:** Md. Ataur Rahman, MD. Hasanur Rahman, A. N. M. Mamun-Or-Rashid, Hongik Hwang, Sooyoung Chung, Bonglee Kim, Hyewhon Rhim

**Affiliations:** 1Department of Pathology, College of Korean Medicine, Kyung Hee University, 26 Kyungheedae-ro, Dongdaemun-gu, Seoul 02447, Korea; hasan079@bsmrstu.edu.bd; 2Korean Medicine-Based Drug Repositioning Cancer Research Center, College of Korean Medicine, Kyung Hee University, 1-5, Hoegidong, Dongdaemungu, Seoul 02447, Korea; 3Global Biotechnology & Biomedical Research Network (GBBRN), Department of Biotechnology and Genetic Engineering, Faculty of Biological Sciences, Islamic University, Kushtia 7003, Bangladesh; 4Anti-Aging Medical Research Center and Glycation Stress Research Center, Graduate School of Life and Medical Sciences, Doshisha University, Kyoto 602-8566, Japan; mamunbtgeiu@gmail.com; 5Center for Neuroscience, Brain Science Institute, Korea Institute of Science and Technology (KIST), 5 Hwarang-ro 14-gil, Seongbuk-gu, Seoul 02792, Korea; hongik.kist@gmail.com (H.H.); sooyoung@kist.re.kr (S.C.); 6Division of Bio-Medical Science and Technology, KIST School, Korea University of Science and Technology (UST), Seoul 02792, Korea

**Keywords:** Alzheimer’s disease (AD), aggregation, autophagy, aggresome, autophagosomes, aggrephagy

## Abstract

Alzheimer’s disease (AD) is one of the most prevailing neurodegenerative diseases in the world, which is characterized by memory dysfunction and the formation of tau and amyloid β (Aβ) aggregates in multiple brain regions, including the hippocampus and cortex. The formation of senile plaques involving tau hyperphosphorylation, fibrillar Aβ, and neurofibrillary tangles (NFTs) is used as a pathological marker of AD and eventually produces aggregation or misfolded protein. Importantly, it has been found that the failure to degrade these aggregate-prone proteins leads to pathological consequences, such as synaptic impairment, cytotoxicity, neuronal atrophy, and memory deficits associated with AD. Recently, increasing evidence has suggested that the autophagy pathway plays a role as a central cellular protection system to prevent the toxicity induced by aggregation or misfolded proteins. Moreover, it has also been revealed that AD-related protein aggresomes could be selectively degraded by autophagosome and lysosomal fusion through the autophagy pathway, which is known as aggrephagy. Therefore, the regulation of autophagy serve as a useful approach to modulate the formation of aggresomes associated with AD. This review focuses on the recent improvements in the application of natural compounds and small molecules as a potential therapeutic approach for AD prevention and treatment via aggrephagy.

## 1. Introduction

Aggresomes are inclusion bodies that consist of aggregated cytoplasmic proteins induced by the overexpression or inhibition of certain proteins of the proteasome system [1], and the accumulation of incorrectly folded proteins is thought to contribute to the etiology of various neurodegenerative diseases [2]. Misfolded protein molecules processing has an important role in maintaining normal cellular function as well as homeostasis. Three protein quality systems have been found to degrade misfolded or aggregated protein, which are ubiquitinated proteasomal degradation, chaperone-mediated degradation, and selective autophagy or aggrephagy [3]. Previous studies have demonstrated that an increasing number of AD-related proteins are associated with aggresomes [4], and non-pathological proteins could form invasive inclusion bodies as well. It is well-known that Aβ production in addition to oligomer accumulation is the key factor of AD pathogenesis [5]. Additionally, neurofibrillary tangles (NFTs) are caused by the neuronal microtubule-stabilizing hyperphosphorylation of tau protein [6]. Particularly, it is mentioned that the impaired degradation of these aggregated proteins molecules causes neuronal atrophy, cytotoxicity, synaptic impairment, memory deficits, and ultimately causes AD pathogenesis [7,8]. It has been hypothesized that autophagy manipulations might be a potentially promising therapeutic strategy target to modulate protein-aggregation-related diseases and toxicity. Recently, it has been revealed that AD-related protein aggregates might be recognized via several receptors which may be degraded through the autophagy–lysosome pathway (ALP); this process is known as the aggrephagy mechanism [4]. However, it has not been determined whether autophagy easily eliminates invasion caused by other proteins associated with AD. In this review, we would like to emphasize the susceptibility of AD-associated proteins to autophagy.

Autophagy, a cellular self-degradation system, can be exploited for the clearance and removal of misfolded protein, cellular constituents, processing antigens, tackling infectious organisms, sequestering additional organelles, suppressing tumors, and controlling cell growth [9,10]. Autophagy supports the clearance of aggregated or misfolded proteins during stress conditions, especially when ubiquitin–proteasome systems and chaperones become overwhelmed [11]. In general, autophagy is thought to be involved in several human diseases, for instance, heart and liver diseases, myopathies, cancer, and neurodegenerative disorders [12]; however, autophagy malfunction is additionally involved in several protein aggregation diseases [13]. Accumulating evidence has indicated that misfolded proteins can impair cell function and viability via a diversity of mechanisms such as proteasome inhibition, pore formation, as well as intracellular transport disruption [14,15]. It has been shown that misfolded protein production exceeds the capability of the ubiquitin–proteasome pathway and molecular chaperone system [16]. However, aggregated or misfolded protein is vigorously sequestered in a microscopically observable way, which forms pericentriolar structures, so-called an aggresomes. These aggresomes are consequently degraded through autophagy, which is generally called the aggresome–autophagy pathway [16,17]. In this study, we reviewed the role of autophagy to modulate aggresomes and their potential therapeutic action for the treatment of AD, focusing on several small molecules which may potentially help improve AD treatment.

## 2. Molecular Mechanism of Alzheimer’s Disease

Alzheimer’s disease (AD) is one of the most prevailing neurodegenerative diseases, characterized by memory dysfunction and the presence of hyperphosphorylated tau and amyloid β (Aβ) aggregates in multiple brain regions, including the hippocampus and cortex [18]. The clinical and pathological appearance of AD comprises memory impairment and slowness while performing normal regular activities [19]. The main pathological features represent intracellular aggregates composed of the extracellular deposition of amyloid-β (Aβ) and phosphorylated tau protein that form neurofibrillary tangles (NFTs) leading to the formation of senile plaques [20,21,22,23] (Figure 1). Aggregated forms of tau protein are reported to appear in the proximal axon. Large, insoluble tau aggregates, which contain irreversibly phosphorylated tau species, do not move in the axons [13]. The production of toxic tau species may damage the transport system in the axon, reduce APP transport towards the synapse, cause APP accumulation in the soma, and thus lead to Aβ aggregation. Neurofibrillary tangles are formed via the hyperphosphorylation of a microtubule-associated protein known as tau, causing it to aggregate, or group, in an insoluble form. The precise mechanism of tangle formation is not completely understood, and it is still controversial as to whether tangles are a primary causative factor in disease or play a more peripheral role. It has been shown that the degree of cognitive impairment in diseases such as AD is significantly correlated with the presence of neurofibrillary tangles. Investigating possible links between tau and NFTs is important for us to better understand processes that lead to the onset of AD at a mechanistic level. Increased Aβ load and NTFs in autophagic vacuoles further exacerbates autophagy stress and AD pathology, which eventually disturbs communication between neurons and their normal function, as well as finally resulting in neuronal death [24]. Synaptic autophagy vacuoles and aggregate clearance failure induce synaptic toxicity which results in diminished synaptic plasticity as a cause of memory dysfunction in AD.

## 3. Molecular Pathway of Autophagy Process

Autophagy can still be triggered to remove the unwanted components of cytoplasm such as protein clusters instead of injured mitochondria [25,26]; however, at times of strain and injury, autophagy mainly remains inside the cell homeostasis [27]. Unc-51-like kinase 1 (ULK1), as well as its start protein complex, is controlled mostly by metabolic sensor systems: mammalian target of rapamycin complex 1 (mTORC1) as a negative controller, and AMP-activated kinase (AMPK), a positive controller of autophagy [28]. The phosphatidylinositol 3-kinase (PI3K) type III nucleation intricate, whose best-known protein is Beclin-1, can then be activated to stimulate the nucleation of a pre-phagophore with the aid of ATG-9-bound vacuoles [29]. The lipidation of LC3-II (ATG-8) protein has a binding site that attaches to LC3-interacting motifs (LIRs) on autophagosome carriers which is responsible for the pre-phagosome [30]. With the help of ATG-4B, ATG-3, and ATG-7, this bonding with phosphatidylethanolamine is carried out with the assistance of an ATG-12-conjugation intricate as well as the phosphatidylinositol 3,4,5-triphosphate (PIP3) binding intricate [31]. Initially, LC3 is SPLIT by ATG-4B to disclose glycine remains formatting LC3-I which is shifted to ATG-7 (E1-similar enzyme which is shared with ATG-12) and is shifted to its particularly E2-like enzyme, ATG-3, and attached to the main group of phosphatidylethanolamines (PE), which generate LC3-II [32,33]. ATG-7 activates ATG-12 and makes the following intermediates with ATG-7 and ATG-10 (E2-similar enzyme) until it is conjugated with ATG-5 [34]: ATG-12/ATG-5 intricates work as E3-similar enzymes to encourage ATG-8-PE entanglement [35]. ATG-16L is needed for the natural process, but it does confirm that the lipidation happens in the correct cell wall. These actions cause the phagophore to enlarge and simply close, which controls the formation of the autophagosome [36]. Just before closing, the autophagosome’s dual layer merges to the lysosomes, allowing the deterioration of the inward membrane and intra-autophagosome cargo to begin [37]. A detailed molecular mechanism of autophagy is presented in Figure 2.

## 4. Mechanism of Aggresome Formation

Aggrephagy is a selective form of autophagy that eliminates aggregated and ubiquitinated proteins [38]. The first stage of aggresome formation involves the accumulation of irregularly folded or unfolded proteins, packed into larger, insoluble aggregates and transported to the microtubule-organizing center (MTOC) [38,39], where aggresomes can cause autophagic degradation. In the autophagic process, autophagosomes undergo lysosomal degradation through dual autophagy membranes encompassing ubiquitin-labeled proteins [20,40]. The autophagy pathway plays an important role in the cytoplasm, but it is not efficient in the nucleus because autophagy-related molecules do not have nuclear inclusions. Aggresomes are membrane-free, microtubule-dependent, cytoplasmic inclusion bodies that form oligomer complexes with amorphous structures containing unfolded or misfolded proteins, which are generally stable and insoluble under physiological conditions [41,42]. Typically, aggregated proteins continue to grow and develop and then oligomerize to form longer insoluble inclusions bodies or aggregates [43]. These aggregates can either link with themselves or join with existing aggresomes located at the MTOC, residing near the cellular centrosome. These aggregates are labeled with ubiquitin alone with heat shock proteins (Hsps), such as Hsp70, and their development in neurodegenerative diseases is demonstrated in Figure 1 [3,44,45,46]. Additionally, aggresomes are primarily surrounded by the cytoskeleton of intermediate filament, and a cage-like structure appears in the cytoskeleton, such as vimentin and keratin, along with neurofilaments [41,47] (Figure 3). This cage-like structure is known to promote structural stability and inhibit non-specific interactions. These aggresomes exert cellular cytotoxicity, but at a later stage, matured bilayer autophagosomes lose their activity [48]. The components of the ubiquitin–proteasome systems are concentrated or grouped into these assemblies before aggregate molecules are discarded through degradation or refolding [49]. Recently, it has been reported that the modulation of aggresome formation and assembly may serve as a novel approach to treat diseases associated with defects in protein conformation.

Histone deacetylase-6 enzyme (HDAC6), important for aggresome formation, is essential for microtubule transport machinery, and misfolded proteins produced from dendrites and axons are shifted towards the move to lysosome-rich MTOC [50]. HDAC6 is considered as the crucial controller of aggresome creation and maintains the central component of autophagy initiation [4]. In aggrephagy, MTOC is packed in aggresomes and degrades via lysosomes [51].

## 5. Clearance of Aggresomes through Autophagy

When the proteasomal degradation system is overwhelmed, the autophagy pathway is activated as an essential cellular defense system to resist incorrect folding and to prevent the accumulation of aggregated proteins [2,52]. In this system, poorly folded and aggregated proteins are selectively identified and transferred to the MTOC (center of microtubule tissue) around the central body via microtubule-based retrograde transport [53]. Accumulating evidence suggests that autophagy not only protects aggressive cells by chelating poorly folded cytotoxic aggregates, but also enriches poorly folded aggregates for subsequent removal [54]. Autophagy is a multi-step progression well characterized by the construction of an insulating membrane, named phagophore, which swells to form the double-membrane autophagosome and fuses with the lysosome to degrade the sequestered cytoplasmic cargo [52,55]. Unlike proteasomes, autophagy does not require substrate expansion and is capable of degrading large protein complexes, protein aggregates, and even entire organelles [56,57]. Moreover, autophagy is a precisely controlled process involving multiple proteins encoded by the Atg (autophagy-related) genes [58,59], and previous studies revealed that autophagy is induced in response to oxidative stress or proteasomal damage and is directly involved in the elimination of aggresomes [60,61].

Emerging evidence suggests that autophagy is responsible for the removal of aggregated and misfolded proteins, and the inhibition of autophagy preferentially affects the degradation of mutant proteins associated with neurodegenerative diseases while leaving their wild-type counterparts unaffected [62]. Although the mechanism of autophagy for the specific elimination of misfolded proteins is not yet clear at the molecular level, the selective isolation of incorrectly folded proteins in aggresomes can facilitate the preferential removal of abnormal autophagy proteins [63]. By chelating the endogenous autophagy inhibitor mTOR (mammalian target of rapamycin) kinase [64], the aggresome can also contribute to the initiation of autophagy [65,66]. Importantly, the conventional concept of autophagy only involves the non-selective removal of misfolded proteins and regular cellular proteins, whereas the aggresome–autophagy pathway is specific to aggregated and misfolded proteins. Therefore, a special type of autophagy induction is thought to be related to the specific elimination of toxicity linked to the aggresome formation. One potential mechanism has been found in parkin-mediated, Lys^63^-linked polyubiquitination which promotes the elimination of misfolded proteins via autophagy, and autophagy occurs by binding to the adapter protein p62 [67], an Ub binding protein interacting with the ubiquitinated proteins via the autophagic mechanical component LC3 through its UBA domain (connected to Ub) and the 22 amino acid LIR (LC3 interaction region) [68]. p62 promotes the binding of a polyubiquitin chain linked to Lys63 [30,31], and the inhibition of its UBA or LIR domain changes the conditions of the ubiquitinated aggregates in the autophagosomes [69,70]. According to recent evidence, p62 may also promote the formation of protein aggregates in addition to promoting the elimination of autophagy [71]. Further studies are required to determine whether p62 is indeed an Ub receptor that regulates the processing of Lys63-linked, folded, polyubiquitinated proteins via an active aggresome–autophagy pathway [72] (Figure 4). As a result, under various stress conditions, an unnecessary level of misfolded proteins is produced in cells as a result of overproduction and UPS damage, resulting in the creation of huge numbers of protein aggregates. p62, a autophagic cargo receptor first identified in mammalian cells, is essential for the development and autophagic removal of intracellular protein aggregates [73]. Biologically, the autophagic elimination of these p62-positive aggregates provides not only the removal of protein aggregates, but also the maintenance of p62’s homeostatic level [74]. This is significant since intracellular p62 production has been linked to the malignant transformation of autophagy-deficient cells [75]. It was recently proposed that, like some other cytoplasmic membraneless compartments, the arranged p62-positive protein aggregates seem to be actually liquid droplets created by phase separation, which is dependent on the relationship between p62 as well as ubiquitin and can also be regulated by p62 post-translational modification [76]. These cytoplasmic aggregates made up of p62 and poly-Ub proteins are also known as p62 inclusion bodies [77]. Therefore, the creation of p62 bodies increases cell survival under nutritional deprivation conditions, in addition to promoting their own breakdown via selective autophagy.

## 6. Molecular Mechanism of the Fusion of Aggresome and Lysosome

In the final stage of degradation, fully matured and closed membrane aggresomes fuse with lysosomes to form hybrid compartment organelles called “autolysosomes” which digest their contents [39] (Figure 2). Although the fusion process is morphologically well characterized, its mechanism remains relatively elusive at the molecular level. Typically, the separation membrane is fused with the lysosomal membrane, and then, the lysosomal hydrolase breaks down the content of the fused (or aggresome/lysosome) autolysosome. The substance (amino acid, etc.) is eventually degraded, and is then either recycled or removed into the extracellular space [78]. Subsequently, the lysosomal fraction of the autolysosomes is recovered to produce new lysosomes [79]. Rab7A (Ypt7 in yeast), mature autophagosomes, HOPS (homologous protein fusion and classification complex), and SNARE receptor (SNAP (soluble adhesion protein NSF)) (NSF, fusion-sensitive) labeled (ethyl-maleimide) proteins are essential for the fusion of autophagosomes and lysosomes [78]. The SNAREs increase the permeability of the membrane, indicating that the membranes open and fuse the content of two adjacent organelles [80]. The Rab7A is transformed into an active form linked to GTP through the action of HOPS (Figure 2). The fusion of organelle binding is dependent on the complex of Rab7A and HOPS (Rab7A downstream effector), which is present in two membranes fused, and SNARE and Rab7A are enriched at the fusion site [81]. SNAREs are also required for the expansion and closure of autophagosomal membranes, the early stages of autophagy, and for the transportation of Atg9 (a transmembrane protein necessary for membrane development) [82]. Recently, it has been found that the SNARE protein syntaxin 17 (STX17) is employed in the outer membrane of autophagosomes and mediates lysosomal fusion [83,84]. The depletion of STX17 prevents the destruction of autolysosomes and leads to the accumulation of autophagosomes in the process of basal and starvation-induced autophagy [85]. It is currently unknown whether STX17 is also involved in selective aggrephagy [86]. Furthermore, when an autophagosomal membrane is formed around the materials to be submerged in the endoplasmic reticulum, STX17 interacts with the autophagosome-labeled Atg14L protein at a very early stage [87,88]. These findings together suggest that STX17 plays an important role at both ends of the autophagy decomposition pathway.

## 7. The Role of Autophagy in Modulating Aggregation in AD

The crucial function of autophagy in preserving neuronal homeostasis is well established in in vivo studies conducted in mice, in which the inhibition of autophagy results in protein deposition and neurodegeneration over time [89]. Decreased autophagy is related to an increase in age-associated neurodegenerative diseases, such as AD [90]. Defective autophagy typically involves the decreased expression of major receptors for autophagy and altered orientation substrate and autophagosome formation. Emphasis has been placed on several aspects, such as the lack of maturation, degradation, and lysosomal alteration [91]. Interestingly, several genes involved in protein homeostasis in neurodegenerative diseases are mutated, such as the lysosome–autophagy system and ubiquitin–proteasome system [92]. In the cases of neurodegenerative diseases, the first marker of altered autophagy is an abnormal quantity of autophagosomes or amphisomes, which can lead to the generation of ROS and other cytotoxic elements. Incidentally, the accumulation of autophagosomes is characterized by an endogenous pool of amyloidogenic Aβ peptides in animal models with AD [93].

The potential function of SQSTM1/p62 in AD progression and other neurodegenerative disorders has been receiving more attention [94]. Compared to the control group, cytoplasmic SQSTM1/p62 protein levels were found to be reduced in the frontal cortex of AD patients [95,96], indicating that SQSTM1/p62 expression is downregulated [96]. Interestingly, oxidative impairment to the promoter of SQSTM1/p62 leads to the reduction in gene transcription, which also appeared in the brain of AD patients [96]. The histopathological analysis of hippocampal and cortical samples from patients with AD showed that the SQSTM1/p62 protein that is contained in inclusion bodies mainly consists of phosphorylated tau, TRAF6 (tumor necrosis factor receptor associated 6), and ubiquitin, promoting the formation of aggresomes [97]. Moreover, the co-localization of SQSTM1/p62 and Keap-1 insoluble deposits was observed in AD brain extracts [98], and the impairment to the promoter of SQSTM1/p62 as well as co-localization of SQSTM1/p62 with protein aggregation have been observed in other neuropathies, such as Huntington’s disease, tauopathy, and α-synucleinopathies [94,96]. Several studies suggested that SQSTM1/p62 plays a central role in the transport of tau protein to proteasome [97], and that there is a negative connection between the levels of SQSTM1/p62 and p-tau [99], indicating that reduced SQSTM1/p62 function likely causes the accumulation of tau positive aggregates with age. Notably, a SQSTM1/p62-deficient mouse model had a typical AD phenotype, including p-tau neurofibrillary tangles, memory impairment, and synaptic depletion [100]. In addition to protein homeostasis, these last two effects are consistent with the assumption that autophagy is necessary for cell remodeling and neuronal plasticity, and autophagy is a prerequisite for neurogenesis and memory processes [101]. However, it is unknown whether a change in the levels of SQSTM1/p62 is correlated with the elimination of abnormal aggresomes in AD patients. Collectively, current findings suggest that it would be promising to decrease the formation of aggresomes in AD via modulating autophagy.

## 8. Potential Therapeutic Action of Autophagy to Control Aggresome Formation in AD Pathogenesis

Natural compounds or small molecules have been used to induce the clearance of aggresomes and to restore or enhance cognitive function in patients with AD [4]. For example, liraglutide has been found to activate the insulin degradation enzyme (IDE), increase cognition function and long-term potentiation (LTP), and reduce Aβ plaque deposition and inflammation in APP/PS1 mice via the mTOR-independent and JNK pathway [102,103]. Additionally, rapamycin, a well-known natural macrolide, was found to mitigate Aβ plaques, liberate cerebral amyloid angiopathy, and enhance memory impairment in the AD mice model of PDAPP, hAPP (J20), and P301S through the inhibition of mTOR activity [104,105]. Moreover, a polyphenolic compound known as curcumin, a PI3K/mTOR inhibitor, not only relieved AD pathology by decreasing Aβ but also repaired spatial memory function in APP/PS1 mice through the degradation of autophagic Aβ aggregates [106]. In clinical trials using JNPL3 mice as an AD model, autophagy induced by methylene blue was shown to decrease the aggregation of insoluble tau through the inhibition of mTOR [107]. Another polyphenol, oleuropein aglycone, also promoted autophagy by releasing Ca^2+^ from the reticulum and preventing the activity of mTOR in TgCRND8 mice, and considerably downregulated Aβ plaque [108]. A carbazole-based fluorophore, SLM, bound to Aβ and prevented Aβ aggregation in 3xTg-AD mouse models by alleviating the pathological and behavioral impairments in AD [109]. Additionally, human prolactin-releasing peptide palmitoylated analog, palm11-PrRP31, was shown to protect neuronal cells in APP/PS1 mice models of AD and decrease astrogliosis, microgliosis, and β-amyloid plaque load as well. [110]. The list of representative natural compounds and small molecules that modulate aggresome formation in AD pathogenesis by enhancing autophagy is summarized in Table 1.

Recently, mTOR-independent transcription factor, TFEB, was shown to reduce aggresome formation and activate autophagy, and several compounds are capable of stimulating TFEB-induced lysosomal biogenesis and mTOR-independent autophagy (Figure 5). For example, ouabain has been found to activate TFEB, decrease the aggregation of p-Tau, and increase cognition function in P301L mice [112]. Cinnamic acid and aspirin were found to activate TFEB promotor and promote lysosomal biogenesis, which decreased the formation of Aβ plaque in a 5 × FAD mice model [113,114]. Trehalose, a disaccharide molecule, activates calcineurin and protein phosphatase-3 CB (PP3CB) by promoting the translocation of TFEB into the nucleus via mTOR-independent autophagy [115]. Additionally, it has been found that, in APP/PS1 mice, trehalose treatment promotes the clearance of Aβ plaque independent of the mTOR pathway [115,116]. Hep14, a cardiac glycoside-ingenol, increased TFEB-induced ALP by activating PKC and inhibiting GSK3β, thus decreasing plaque formation in an APP/PS1 AD mice model [117]. Jiang et al. demonstrated that the treatment of temsirolimus successfully improved the autophagic clearance of hyperphosphorylated tau in the brain of P301S transgenic mice and okadaic-acid-incubated SH-SY5Y cells. In addition, temsirolimus administration improved memory impairments and spatial learning function in P301S mice [118]. Small-molecule or natural compounds may control autophagy–lysosomal process via mTOR- and TFEB-mediated pathways, which is summarized in Figure 3.

## 9. Future Prospective of Inhibiting Aggresome Formation as a Treatment for AD

Recently, extensive resources were dedicated to the development of high-performance, automated detection platforms to identify compounds that can prevent aggresome development and endorse aggresome formation [119]. Such high-performance screening primarily uses large libraries designed for general-purpose detection, which serves as a useful approach when running screens with unknown targets or with no structural information available [119,120]. These screenings play an important role in drug discovery and the creation of new forms of chemical treatment [119,120]. However, since protein aggregation and the formation of aggresomes are complex and mediated by multi-step processes [14], special care must be taken in interpreting the results, especially when aggresome-related genes are used as reporter genes. For example, inhibiting the early stage of the autophagy pathway reduces the formation of aggresomes and the levels of toxic protein species [121] (Figure 6). However, although preventing the late stages of the autophagic pathway reduces aggresome development, it was shown to increase the assembly of soluble, toxic protein species. This complexity highlights the importance of verifying and characterizing the target sites as well as working mechanisms of candidate compounds. The familial form of Alzheimer’s disease (FAD), resulting from mutations in the PRESENILIN 1 (PSEN1) as well as APP genes, accounts for less than 5% of all AD cases and manifests itself early [122]. An iPSC line generated from a FAD patient with the PSEN1-G206D mutation was created and characterized. The iPSC line kept its original genotype as well as karyotype, was free of Sendai viral vectors and reprogramming factors (OCT4, c-MYC, SOX2, and KLF4), had a typical morphology, expressed endogenous pluripotency markers, and could be differentiated into ectodermal, mesodermal, as well as endodermal cells, clarifying its pluripotency [123,124]. PSEN1 mutations cause severe cognitive deterioration, which is largely caused by Aβ oligomerization and accumulation, as well as Tau phosphorylation [125]. The iPSC line revealed here will allow researchers to investigate the involvement of Aβ, p-Tau, and other pathways in FAD, paving the way for new treatments for the condition [126]. Therefore, patient-specific iPSCs include defective genes and may be an ideal cell model for studying the pathophysiology of AD, aiding in the medication screening of AD treatment.

## 10. Conclusions

Autophagy enhancers that can be utilized for the potential treatment of neurodegenerative diseases are receiving growing attention in recent studies [20,127,128,129]. It has been shown that the inactivation of mTOR via the lipophilic macrolide antibiotic rapamycin promotes the induction of autophagy [130]. Furthermore, the long-term administration of rapamycin can reduce amyloid loads in mouse models of AD and improve the cognitive function as well as the pathology of tauopathy [131,132]. In addition, post-translational modifications, such as ubiquitination, acetylation, O-GlcNAcylation, and phosphorylation, also appear to exert a positive effect on autophagy [70,133]. Acetylation is an important cellular mechanism that protects cells from stress stimuli and can be changed in neurodegenerative pathologies, highlighting acetylation and deacetylation processes as potential therapeutic candidates for neurodegenerative diseases. Several selective, small-molecule inhibitors of HDAC6 have been identified [134]. For example, tubastatin A promotes the acetylation of α-tubulin, stabilizes the microtubule network, and confers neuroprotection on neurons during in vitro culture and neurodegeneration. By clarifying the mechanism underlying the selection of lysosomal loads, an autophagy-based approach will serve as a more effective therapeutic candidate [135]. In the case of AD, aggresome formation can occur, which can induce the inhibition of proteasome activity. Aggrephagy and its downstream signaling cascades offer promising new therapeutic targets for preventing AD, and further research is required to clarify the relationship between the mechanisms involved in autophagic activities and the formation of aggresomes.

## Figures and Tables

**Figure 1 biomedicines-10-01027-f001:**
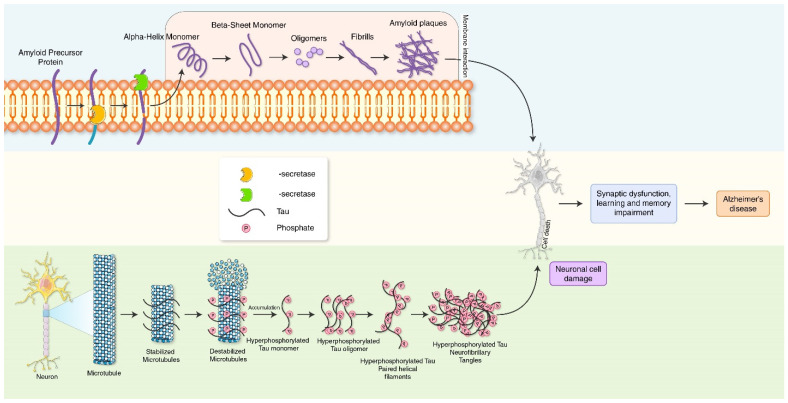
Molecular mechanism of AD pathogenesis. Amyloid precursor protein cleaves by β-secretase and α-secretase, leading to the accumulation of neurotoxic Aβ plague. Neurofibrillary tangles (NFTs) are aggregates of hyperphosphorylated tau protein. Aβ plague and NFTs causes synaptic dysfunction and memory impairment, which are characteristics of AD.

**Figure 2 biomedicines-10-01027-f002:**
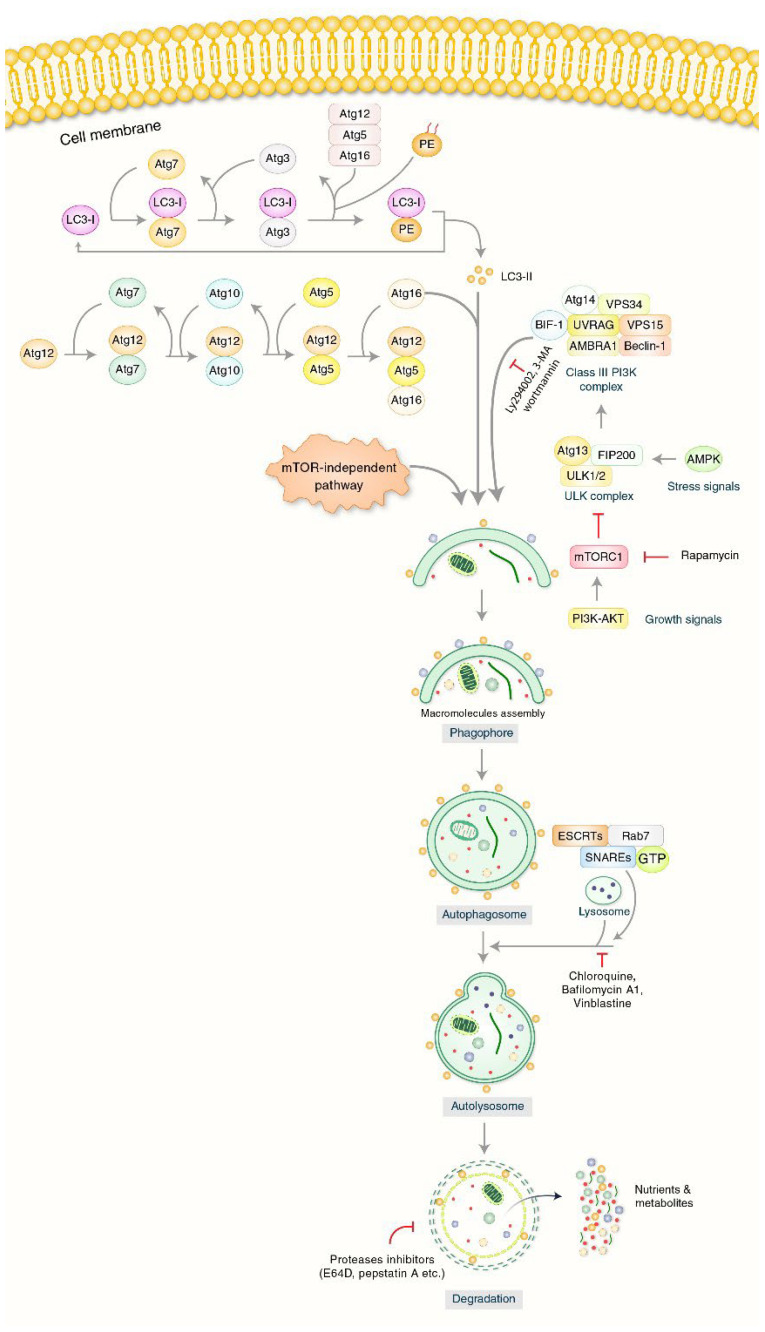
Molecular mechanism of autophagy. Autophagy process is initiated through the formation of the phagophore structure formation. PI3K-AKT and mTOR have been predisposed to form phagophore initiation. Beclin-1, UVRAG, and VPS34 complexes help to initiate phagophore formation. Phagophore nucleation has been extended to autophagosome formation. Binding between mature autophagosomes as well as lysosomes indicates the autolysosome formation. Chloroquine (CQ), Bafilomycin A1 (BAF-A1), and Vinblastine inhibit the binding of lysosomes and autophagosomes. Eventually, autolysosomes will be eliminated by acid hydrolases that produce nutrients as well as recycling metabolites.

**Figure 3 biomedicines-10-01027-f003:**
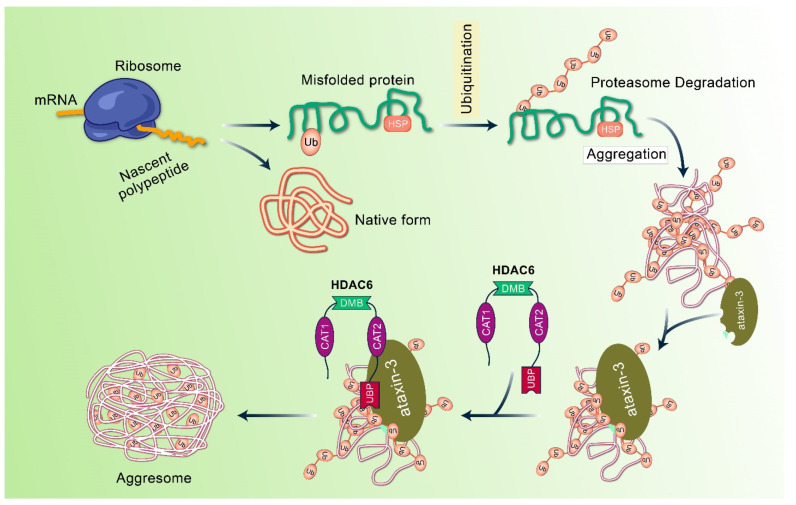
Molecular mechanism of aggresome development and formation. Under normal conditions, misfolded and polyubiquitinated proteins are fragmented via the ubiquitin proteasomal system. When ubiquitin proteasomal system is altered or overhauled, misfolded polyubiquitin proteins accumulate and form to aggregate. In this case, ataxin-3-dexiquitinase interrelates and aggregates with polyubiquitinated proteins to form ubiquitin chain structure. In addition, HDAC6 binds these non-anchored C-terminal tails of ubiquitin to form aggregates and recruits them into the dynein motor complex.

**Figure 4 biomedicines-10-01027-f004:**
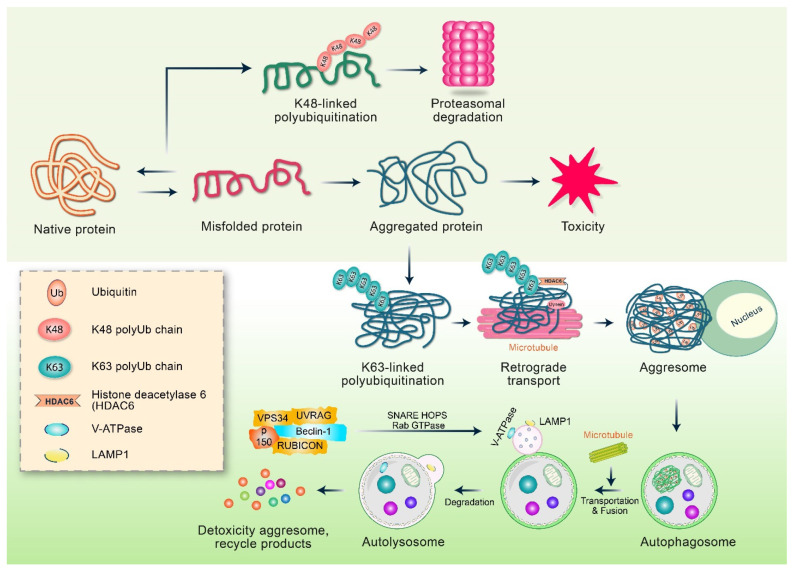
The regulation of aggresome–autophagy pathway. Aggresome and autophagy are regulated by Lys63-associated polyquitination-mediated pathway. Oxidative damage or genetic mutations are responsible for protein misfolding. After folding, the misfolded proteins are labeled with polyubiquitin chains linked to Lys48, which is subsequently degraded via either the proteasomal system or chaperone-mediated pathway. However, when proteasome and chaperone systems are overwhelmed, misfolded proteins form oligomers or aggregates with cellular toxicity. Moreover, PD-related parkin ligase E3 acts with the E2 enzymes Ubc13/Uev1a to facilitate Lys63-associated polyubiquitination of misfolded proteins in proteasome injury conditions. Polyubiquitin chain stimulates binding with HDAC6, and the misfolded proteins bind to dynein motor complex, which retrograde the aggresomes to MTOC via transport. Polyubiquitination encourages p62 binding in addition to recruiting autophagic membranes to form autophagosomes. Consequently, the fusion of autophagosomes and lysosomes facilitates the degradation of misfolded and aggregated proteins through the lysosomal hydrolases.

**Figure 5 biomedicines-10-01027-f005:**
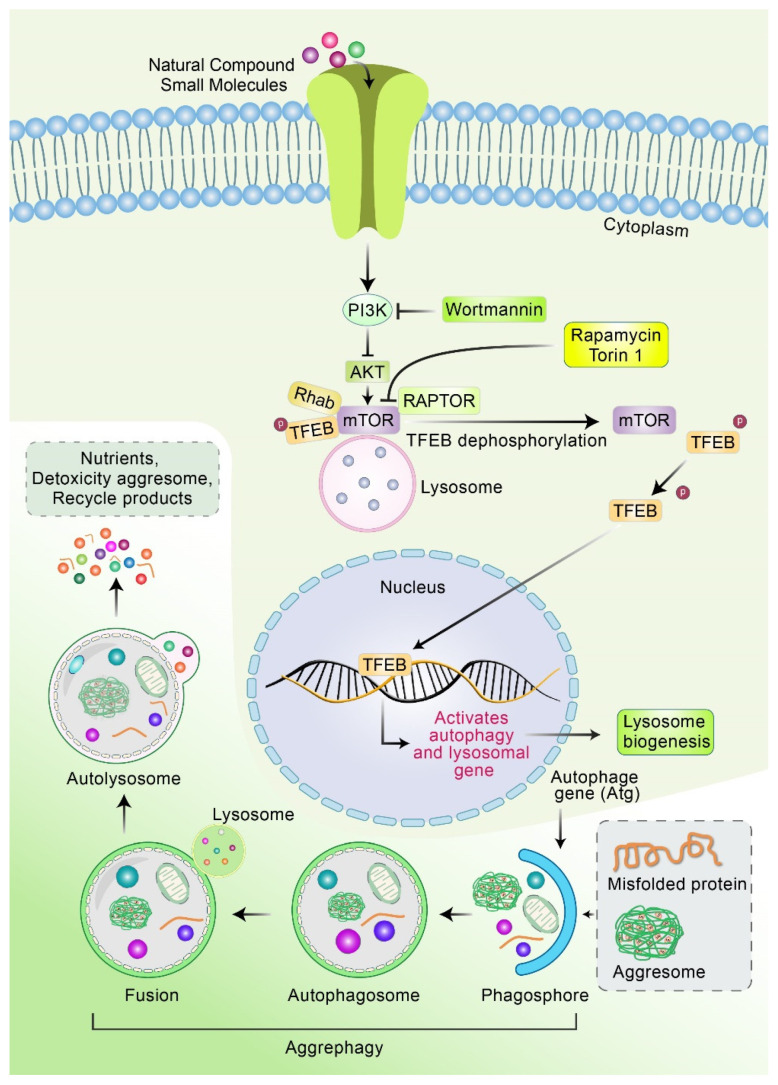
Mechanism of natural compounds or small molecules to activate mTOR and TFEB in autophagy–lysosomal process. Natural compounds or small molecules inactivate AKT and mTOR, which promote the accumulation of TFEB in the cytoplasmic and its nuclear translocation. TFEB in cytoplasm is heavily phosphorylated and interacts with mTOR in the lysosome surface. The inactivation of mTOR activity stimulates the dephosphorylation of TFEB. Subsequently, dephosphorylated TFEB is translocated from cytoplasm to nucleus. In nucleus, TFEB binds to the promoter regions of autophagy- and lysosomal-associated genes and induces gene expression in addition to lysosome biogenesis. The aggresome is bound to the phagophore, resulting in the formation of the autophagosome. Eventually, the autophagosome fused with lysosome degrades aggresomes via aggrephagy process.

**Figure 6 biomedicines-10-01027-f006:**
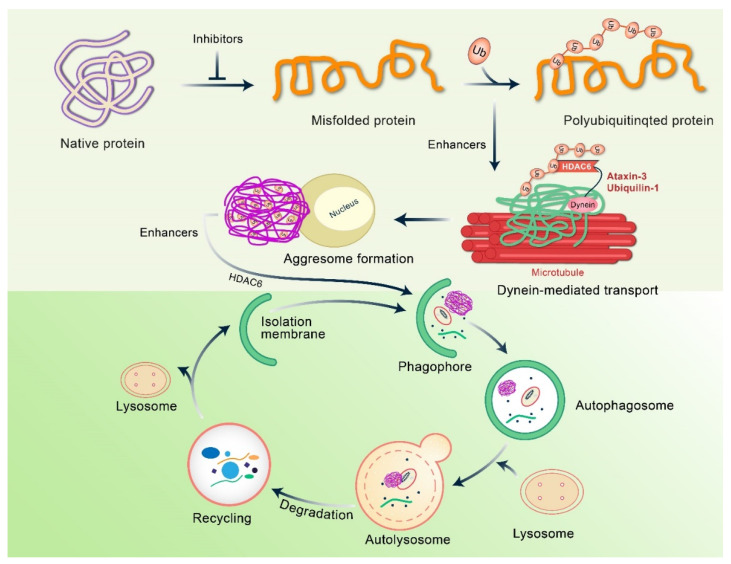
Regulation of misfolded protein by autophagy. Misfolded proteins are recognized and polyubiquitinated by ubiquitin E3 protein ligases. Adapter proteins, such as HDAC6, ataxin3, and ubiquitin-1, bind to polyubiquitinated proteins on the dynein motor complex for retrograde transport to the aggressor. Aggresome takes over the autophagy mechanism, including HDAC6, and breaks down aggresome. Several steps along this path enable small molecules to block incorrect protein folding and improve the coupling of folded proteins to dynein for retrograde transport, or improve the clearance of aggresome by autophagy, which can be potentially applied in the treatment of neurodegenerative diseases.

**Table 1 biomedicines-10-01027-t001:** Natural compounds and small molecules that modulate autophagic activity and control aggresome formation in AD models.

Natural Compounds/Small Molecules	AD Model	Molecular Mechanism	Research Outcomes	References
Fisetin	Mouse and rat primary cortical neurons	mTOR inhibition, TFEB and Nrf2 activation	Autophagy induction, decreases sarkosyl-insoluble tau phosphorylation	[111]
Ouabain	Tau transgenic fly, P301L mice	Inactivation of mTOR, activation of TFEB	Increases autophagy, decreases toxic tau, increases memory function	[112]
SLM, a carbazole-based fluorophore	3xTg-AD	Activation of GSK-3β, reduces neuroinflammation	Decreases Aβ40 and Aβ42 levels, reduces phosphorylation of tau	[109]
Aspirin	5xFAD	Activation of PPARα and TFEB	Increases lysosomal biogenesis, decreases Aβ	[113]
Liraglutide	APP/PS1, APPswe/SH-SY5Y cells	Increase in IDE levels, mTOR-independent, JNK activation	Improves cognitive function, reduces Aβ plaque deposition and inflammation, enhances LTP and autophagy activation	[102,103]
Rapamycin	Transgenic (h)APP mice	mTOR inactivation	Improves memory, decreases sarkosyl-insoluble tau	[104,105]
Cinnamic acid	5 × FAD	Activation of PPARα, upregulation of TFEB	Reduces cerebral Aβ plaque burden, improves memory function, stimulates lysosomal biogenesis.	[114]
Trehalose	APP/PS1, Tg2576	Increase in synaptophysin, doublecortin, and progranulin	Inhibits tau, improves cognitive and learning ability	[115,116]
Curcumin	APP/PS1	mTOR inactivation	Reduces Aβ plaque, increases memory function	[106]
Oleuropein aglycone	TgCRND8 mice	Inhibition of mTOR and Ca^2+^ liberating	Reduces Aβ plaque, increases synaptic plasticity	[108]
Hep-14	APP/PS1	Upregulation of TFEB	Reduces Aβ plaque	[117]
Palm^11^-PrRP31	APP/PS1	Activation of pre-synaptic marker synaptophysin	Decreases Tau phosphorylation, reduces Aβ plaque and microgliosis	[110]
Methylene blue	JNPL3	mTOR inactivation	reduces insoluble tau, increases memory function	[107]
Temsirolimus	P301S mice	Inhibition of mTOR	Improves motor and memory function, reduces sarkosyl-insoluble tau	[118]

## Data Availability

Request upon corresponding author.

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
