# Peer review of "Autophagy Modulation in Aggresome Formation: Emerging Implications and Treatments of Alzheimer’s Disease"

_biomedicines, 2022, doi:10.3390/biomedicines10051027_

Round 1

Reviewer 1 Report

The article titled “Autophagy modulation in aggresome formation: ” by Rahman et al. is a well written and updated review on the state-of-the-art of relevant aspects of autophagy, highlighting the importance of aggresome formation and aggrephagy in Alzheimer´s disease (AD) neurodegeneration and on its potential treatment. I have minor comments and requests, as follows:

  1. Figure 1: Please, add the words “hyperphosphorylated Tau “, and make the P group a bit bigger.
  2. Figure 2: It should be clearly indicated where is the phagophore membrane coming from.
  3.  
  4. Page 7: The role of p62 in the formation of protein aggregates in addition to promoting the elimination of autophagy should be elaborated with some more detail.
  5. Figure 5: Please, check the spelling in this sentence written within the figure: “activate autophagy and lysosome gene “
  6. The authors should discuss recent articles stressing the importance of the induced pluripotent stem cell (iPSCs) technology to the field of AD research, including (among others):

-Gain of toxic apolipoprotein E4 effects in human iPSC-derived neurons is ameliorated by a small-molecule structure corrector.

Nature Medicine 2018 May; 24(5): 647–657.

- Generation of an integration-free iPSC line, ICCSICi006-A, derived from a male

Alzheimer's disease patient carrying the PSEN1-G206D mutation.

Stem Cell Research, 2019, 40, 101574

Nature Medicine 2018 May; 24(5): 647–657.

- Generation of an integration-free iPSC line, ICCSICi006-A, derived from a male

Alzheimer's disease patient carrying the PSEN1-G206D mutation.

Stem Cell Research, 2019, 40, 101574

Author Response

The article titled “Autophagy modulation in aggresome formation: ” by Rahman et al. is a well written and updated review on the state-of-the-art of relevant aspects of autophagy, highlighting the importance of aggresome formation and aggrephagy in Alzheimer´s disease (AD) neurodegeneration and on its potential treatment. I have minor comments and requests, as follows:

  1. Figure 1: Please, add the words “hyperphosphorylated Tau “, and make the P group a bit bigger.

>> (Response) We modified figure 1 accordingly (page 5).

  1. Figure 2: It should be clearly indicated where is the phagophore membrane coming from.

>> (Response) We added phagophore membrane where it is coming from in figure 2 (page 7).

  1. Page 7: The role of p62 in the formation of protein aggregates in addition to promoting the elimination of autophagy should be elaborated with some more detail.

>> (Response) We added role of p62 in the formation of protein aggregates in addition to promoting the elimination of autophagy (page 10 & 11, line 224-238).

  1. Figure 5: Please, check the spelling in this sentence written within the figure: “activate autophagy and lysosome gene “

>> (Response) We checked and modified figure 5 (page 18).

  1. The authors should discuss recent articles stressing the importance of the induced pluripotent stem cell (iPSCs) technology to the field of AD research, including (among others):

-Gain of toxic apolipoprotein E4 effects in human iPSC-derived neurons is ameliorated by a small-molecule structure corrector.

Nature Medicine 2018 May; 24(5): 647–657.

- Generation of an integration-free iPSC line, ICCSICi006-A, derived from a male

Alzheimer's disease patient carrying the PSEN1-G206D mutation.

Stem Cell Research, 2019, 40, 101574

>> (Response) We added the importance of the induced pluripotent stem cell (iPSCs) technology to the field of AD research as the mention paper and used as references in section 9. ‘Future prospective of inhibiting aggresome formation as a treatment for AD’ (page 19, line 383-396).

Reviewer 2 Report

A very good paper. Congratulations. however, extensive editing is needed for the increase in readability.

I have enclosed a marked paper with several changes to be made in the language.

Otherwise the paper is very good, exhausitive and very informative

Author Response

A very good paper. Congratulations. however, extensive editing is needed for the increase in readability. I have enclosed a marked paper with several changes to be made in the language. Otherwise the paper is very good, exhausitive and very informative

>> (Response) First of all, we would like to express our sincere gratitude for the time and effort the reviewer had put into reviewing our manuscript. We have incorporated changes based on the reviewer comments provided in the manuscript which revised parts are highlighted by BLUE color in the entire revised manuscript.

Reviewer 3 Report

In this review, authors focused on the role of autophagy to modulate aggresome and theri potential action for the treatment of AD focusing on several smal molecules autophagy modulators which may potentially aid to improve AD treatment. It is a very complete and exhaustive review, well written, with recent bibliography, with very clear and informative figures/schemas. I consider that it may suitable for published.

Author Response

In this review, authors focused on the role of autophagy to modulate aggresome and theri potential action for the treatment of AD focusing on several smal molecules autophagy modulators which may potentially aid to improve AD treatment. It is a very complete and exhaustive review, well written, with recent bibliography, with very clear and informative figures/schemas. I consider that it may suitable for published.

>> (Response) First of all, we would like to express our sincere gratitude for the time and effort the reviewer had put into reviewing our manuscript. We are grateful to reviewer positive comments.